

# Circulation changes in the Amundsen Basin from 1991 to 2015 revealed from distributions of dissolved [230]Th

Ole Valk[1], Michiel M. Rutgers van der Loeff[1], Walter Geibert[1], Sandra Gdaniec[2], S. Bradley Moran[3], Kate Lepore[4], Robert Lawrence Edwards[5], Yanbin Lu[6], Viena Puigcorbé[7], Nuria Casacuberta[8,9], Ronja
Paffrath[10] William Smethie[11], Matthieu Roy-Barman[12]

[1]Alfred Wegener Institute Helmholtz Centre for Polar and Marine Research, 27570 Bremerhaven, Germany
[2]Stockholm University, Department of Geological Sciences, 106 91, Stockholm, Sweden
[3]College of Fisheries and Ocean Sciences, University of Alaska Fairbanks, Fairbanks, AK 99775, USA
[4]Mount Holyoke College, South Hadley, MA 01075, USA
[5]University of Minnesota, Minneapolis, MN 55455, USA
[6]Nanyang Technological University, 639798, Singapore
[7]Center for Marine Ecosystem Research, School of Science, Edith Cowan University, Joondalup, WA 6027, Australia
[8]Laboratory of Ion Beam Physics, ETH Zurich, 8093 Zurich, Switzerland
[9]Institute of Biogeochemistry and Pollutant Dynamics, Environmental Physics, ETH Zurich, 8092 Zurich, Switzerland
[10]Max Planck Research Group for Marine Isotope Geochemistry, Institute for Chemistry and Biology of the Marine Environment, University of Oldenburg, 26129, Oldenburg, Germany
[11]Lamont-Doherty Earth Observatory, Palisades, NY 10964-8000, USA
[12]Laboratoire des Sciences du Climat et de l'Environnement, LSCE/IPSL, CEA – CNRS – UVSQ, Université Paris-Saclay, 91191 Gif-sur-Yvette, France

*Correspondence to*: Ole Valk (ole.valk@awi.de)

**Abstract.** This study provides dissolved and particulate [230]Th and [232]Th results as well as particulate [234]Th data collected during expeditions to the central Arctic Ocean on ARK-XXIX/3 (2015) and ARK-XXII/2 (2007) (GEOTRACES sections GN04 and GIPY11, respectively). Constructing a time-series of dissolved [230]Th from 1991 to 2015 enables the identification of processes that control the temporal development of [230]Th distributions in the Amundsen Basin. After 2007, [230]Th concentrations
decreased significantly over the entire water column, particularly between 300 m and 1500 m. This decrease is accompanied by a circulation change, evidenced by a concomitant increase in salinity. Potentially increased inflow of water of Atlantic origin with low dissolved [230]Th concentrations leads to the observed depletion in dissolved [230]Th in the central Arctic. Because atmospherically derived tracers (CFC, [3]He/[3]H) do not reveal an increase in ventilation rate, it is suggested that these interior waters have undergone enhanced scavenging of Th during transit from the Fram Strait and the Barents Sea to the central
Amundsen Basin. The [230]Th depletion propagates downward in the water column by settling particles and reversible scavenging. Taken together, the temporal evolution of Th distributions point to significant changes in the large-scale circulation of the Amundsen Basin.





## 1 Introduction

The Arctic Ocean is one of the most rapidly changing parts of the Earth's ocean-atmosphere system as a result of climate change. Underlying the potential anthropogenic changes is a large natural variability of the Arctic. Due to the limited observations in this extreme environment, establishing datasets that allow an assessment of its variability is important. Natural
tracers of physical, chemical and biological processes provide an integrated description of the changing state of the system. They are therefore key tools to investigate processes, monitor environmental changes, and provide an observational baseline against which models can be tested.

### 1.1 Hydrography and Circulation patters of the central Arctic Ocean

The central Arctic Ocean is divided into the Amerasian Basin and Eurasian Basin by the Lomonosov Ridge (Fig. 1). The
Gakkel Ridge separates the Eurasian Basin further into the Nansen Basin and the Amundsen Basin, while the Amerasian Basin is separated into the Makarov and Canada Basin by the Alpha-Mendeleev Ridge.

Water masses of the Arctic Ocean are commonly distinguished as five layers (Rudels, 2009). The uppermost low salinity Polar Mixed Layer (PML) varies in thickness between winter and summer, due to melting and freezing of sea ice. Salinity ranges from 30 to 32.5 (Amerasian Basin) to 32-34 (Eurasian Basin). Below the PML is a 100–250-m-thick halocline in which salinity
increases sharply from approximately 32.5 to 34.5. The underlying Atlantic Layer is characterized in salinity and temperature by waters of Atlantic origin and is usually found between 400 m and 700 m water depth. Its salinity is 34.5-35. Intermediate waters down to 1500 m, with a salinity of 34.87-34.92, are still able to exchange over the Lomonosov Ridge. In contrast, deep and bottom waters differ between the Eurasian Basin (salinity: 34.92-34.945) and the Amerasian Basin (salinity: 34.92-34.96) due to the topographic barrier.
Atlantic waters from the Norwegian Atlantic Current enters the Arctic Ocean via the Fram Strait and the Barents Sea. Fram Strait Branch Water (FSBW) is supplied through the West Spitsbergen Current (WSC) (Rudels et al., 2012) (Fig. 1). Barents Sea Branch Water (BSBW) enters through the Barents Sea and consists of Atlantic water that undergoes strong modifications in Barents- and Kara Seas by cooling down and mixing with continental runoff and meltwater (Rudels et al., 2015). The BSBW enters the Nansen Basin through the Santa Anna Trough, where limited mixing with the FSBW occurs. Once in the polar
ocean, surface waters follow wind driven ice motion (Aagaard et al., 1980), whereas deeper Atlantic water branches (FSBW and BSBW) flow cyclonically to the east forming a boundary current along the continental slopes of the Nansen and Amundsen basins.

BSBW (around approx. 1025 m depth, Tanhua, 2009) and FSBW (approx. 425 m) return in the Atlantic and Intermediate water layers along the Lomonosov Ridge towards Fram Strait (Rudels et al., 2013) (Fig. 1) and a second branch crosses the
Lomonosov Ridge entering the Canada Basin following the Arctic Ocean Boundary Current (AOBC) (Rudels, 2009).

Deep waters of the Arctic Ocean have similar structure, with a thick intermediate layer stratified in temperature but with salinity almost constant with depth (Rudels, 2009). Yet, the Amerasian Basin deep water is warmer, saltier and less dense than



the Eurasian Basin Deep Water (EBDW) (Aagaard, 1981;Worthington, 1953). The deepest exchange of Makarov Basin water, part of the Amerasian Basin, occurs through a depression of the ridge, called the Intra-Basin with sill depth of approximately 1800 m (Björk et al., 2007; Jones et al., 1995; Björk et al., 2010). Water from the Amundsen Basin flows over the Lomonosov Ridge into the deep Makarov Basin and in the reverse direction through this pathway (Middag et al., 2009).

the Lomonosov Ridge into the deep Makarov Basin and in the reverse direction through this pathway (Middag et al., 2009). Another important component of the Arctic Ocean is the freshwater content, coming from the melting of sea-ice and from river runoff. The fresh water content of the central Arctic Ocean is currently at the highest level since the early 1980s, and is expected to increase in the future (Rabe et al., 2014) which could lead to a stronger stratification of the water column. This process is supported by sea ice decline, as observed in the Beaufort Gyre (Wang et al., 2018). Karcher et al. (2012) suggest a reversal in

flow direction of Atlantic Water in the Canada Basin at intermediate water depths on basis of [129]I observations and modelling. This could lead to a decoupling of flow regimes in the Canada and Eurasian Basins and reduce exchange times between the two major basins of the Arctic Ocean (Karcher et al., 2012).

**1.2 Particle Fluxes, shelf input and biological productivity**

Biological productivity in the central Arctic Ocean and related particle fluxes are lower than in other oceans due to the perennial

sea ice cover (Clark and Hanson, 1983). This is expected to change in the future when light limitation is relieved by sea ice retreat (Pabi et al., 2008). Arctic sea-ice extent is declining (Serreze et al., 2016) and ice is becoming thinner (Serreze and Stroeve, 2015). Biological productivity may increase and begin earlier in the year, at least in the Pacific part of the Arctic, depending on nutrient supply (Hill et al., 2017). Recent studies show that productivity is still low in the central Arctic Ocean, limited by both light and nutrient availability (Arrigo and van Dijken, 2015). Highest net community production (NCP) is

found at the ice edge of the Nansen Basin and over the shelves, while the Amundsen Basin shows the lowest NCP (Ulfsbo et al., 2014). Apart from the possible effect on NCP, the declining sea-ice cover will also enhance ice derived particle fluxes (Arrigo et al., 2008; Boetius et al., 2013). The Arctic Ocean has the largest relative amount of shelves of all World Ocean, approximately 30% of area in total. Shelf sediments and large volumes of riverine input add trace metals and carbon among other terrestrial components to Arctic shelf areas, some of which are transported to the central Arctic by the Transpolar Drift

(TPD) (Wheeler et al., 1997; Rutgers van der Loeff et al., 2018; Rutgers van der Loeff et al., 1995). On the basis of an increase of [228]Ra supply to the interior Arctic Ocean, Kipp et al. (2018) suggested that the supply of shelf derived materials is increasing with a following change in trace metal, nutrient and carbon balances. Thawing permafrost and subsequent increasing coastal erosion (Günther et al., 2013) may increase terrestrial input to the central Arctic Ocean (Schuur et al., 2013; Schuur et al., 2015).

**1.3 Th as a tracer of water circulation and particle fluxes**

Thorium isotopes have been extensively used to study and model physical oceanographic processes, such as advection, water mass mixing and particle flux (Bacon and Anderson, 1982; Rutgers van der Loeff and Berger, 1993; Roy-Barman, 2009;





Rempfer et al., 2017). In seawater, $^{230}$Th ($t_{1/2}$=75380 yrs) is produced by the radioactive decay of dissolved $^{234}$U. Without lateral transport by currents, the vertical distribution of $^{230}$Th in the water column is controlled by reversible exchange with sinking particles and increases with depth (Bacon and Anderson, 1982; Nozaki et al., 1981). Deviations from a linear increase with depth profile of $^{230}$Th (Bacon and Anderson, 1982) suggest that oceanic currents transport $^{230}$Th away from the production

area, or that ventilation, upwelling, or depth-dependent scavenging processes play a role for the $^{230}$Th distribution in the water column (e.g., Rutgers van der Loeff and Berger, 1993; Moran et al. 1995; Roy-Barman, 2009).

$^{232}$Th is known as a tracer for shelf/continental derived signatures (Hsieh et al., 2011), while $^{234}$Th serves as a tracer for particle flux (Moran and Smith, 2000).

### 1.3.1 $^{230}$Th in the Arctic Ocean

Several studies have addressed $^{230}$Th in the Arctic Ocean over the past decades. Yet several key points to understand removal processes of dissolved $^{230}$Th are not entirely understood and the sensitivity of dissolved $^{230}$Th to environmental changes is still not explained sufficiently.

Bacon et al. (1989) reported the first study of $^{230}$Th and $^{231}$Pa in the Arctic in 1983 at CESAR Ice Camp, located at the Alpha Ridge. They hypothesized that scavenging of reactive elements in the central Arctic Ocean was significantly lower than in

other parts of the world to explain the high $^{230}$Th concentrations observed at the Alpha Ridge and the northern Makarov Basin (Bacon et al., 1989).

Cochran et al. (1995) presented the first $^{230}$Th study for the Eurasian Basin. They showed that deep water in the central Nansen Basin has lower particulate and higher dissolved $^{230}$Th concentrations than near the slopes (Cochran et al., 1995).

Dissolved $^{230}$Th concentrations in the Nansen Basin were lower than those from the Alpha Ridge reported by Bacon et al.

(1989). Residence times of dissolved $^{230}$Th were calculated to be 18-19 years in the central Nansen Basin and 10-12 years on the Barents Sea slope (Cochran et al., 1995).

Scholten et al. (1995) reported $^{230}$Th concentrations in the Nansen, Amundsen, and Makarov Basins. They found that the shallower EBDW is influenced by ventilation, in contrast to the deeper Eurasian Basin Bottom Water (EBBW). They suggested resuspension as the cause for the increased scavenging rates in the EBBW.

Edmonds et al. (1998), later confirmed by Trimble et al. (2004), showed that $^{230}$Th activities in the deep southern Canada Basin were much lower, and residence times correspondingly shorter, than observed by Bacon et al. (1989) at the Alpha Ridge.

Moran et al. (2005) reported surface sediment $^{231}$Pa$_{xs}$/$^{230}$Th$_{xs}$ from the Canada Basin. They provided new insights into the relevance of scavenging removal and the horizontal redistribution of these tracers as well as the fractionation between the low productivity, sea ice covered interior basins and the seasonally high particle flux areas at the margins. Low surface sediment

$^{231}$Pa$_{xs}$/$^{230}$Th$_{xs}$ ratios were interpreted as a result of chemical fractionation of $^{230}$Th and $^{231}$Pa in the water column resulting in preferred $^{231}$Pa export out of the Arctic. Almost all of the $^{230}$Th produced in-situ (ca. 90 %) was estimated to be removed within the Arctic by scavenging onto particles (Moran et al., 2005).

Roy-Barman (2009) presented a boundary scavenging profile model, showing that linear $^{230}$Th concentration profiles do not necessarily imply that circulation is negligible. They suggested that the difference between the Arctic and other oceans is a considerable lateral transport of $^{230}$Th from the interior to the margins.

Hoffmann et al. (2013) presented new $^{231}$Pa$_{xs}$/$^{230}$Th$_{xs}$ data in well-dated sediment cores and suggested that the deep waters of

the Arctic is exchanged through the Fram Strait on centennial timescales.

Valk et al. (2018) showed that the deep Nansen Basin is influenced by volcanic and hydrothermal inputs that lead to scavenging removal of $^{230}$Th over several years, at least episodically.

This overview shows that the regional distribution of dissolved $^{230}$Th in relation to particle fluxes and water mass residence time is known to a certain degree, but the knowledge about temporal development of this tracer and the connected processes

is still very limited..

## 1.4 Motivation

Global warming is triggering profound changes in the ocean, and the Arctic Ocean is especially vulnerable to such environmental forcing. Summer ice cover is rapidly declining, as are changes in the supply of terrestrial material (Günther et al., 2013), particle flux (Boetius et al., 2013) and ocean circulation (Karcher et al., 2012). These developments are expected to

leave an imprint on the distribution of particle-reactive radionuclides, such as Th isotopes. A central motivation for this GEOTRACES study is to use the Th isotopes to depict changes in circulation and particle fluxes in the Arctic Ocean from 1991 to 2015. The basis of this study is a time series consisting of natural radionuclide data from various previous studies combined with new data from 2007 and 2015.

## 2 Methods

### 2.1 Sampling and analysis of Th in samples collected in 2007

Sea water samples were filtered directly from the 24 L CTD-Niskin® bottles into acid cleaned cubitainers (LDPE) using 0.45 µm pore size Acropaks®. Samples were collected in volumes of 1 L, 2 L, and 10 L and acidified with concentrated ultraclean HNO₃. Samples for the analysis of total $^{230}$Th were taken without filtration. Analyses were performed at the University of Minnesota, Minneapolis, following methods from Shen et al. (2003). Measurements were done using Inductively Coupled

Plasma Mass Spectrometry (ICP-MS, Thermo Finnigan, Neptune) equipped with a Secondary Electron Multiplier (SEM) and a Retarding Potential Quadrupole (RPQ) energy filter.

### 2.2 Sampling and analysis of dissolved Th samples collected in 2015

Samples were filtered directly from the 24 L CTD-Niskin® bottles into cubitainers (LDPE) through 0.45 µm pore size Acropaks® in volumes of 10 L (>2000 m) and 20 L (<2000 m), according to the expected concentrations (Nozaki et al., 1981).



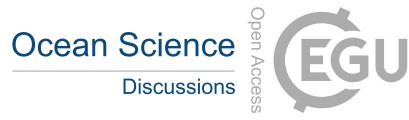

Acropaks® were used for half of the cruise and then replaced by new ones. Subsequently water samples were acidified to a pH of 1.5-2 by addition of 1 mL (acid)/L (seawater) of concentrated double distilled $HNO_3$.

Preconcentration and analysis of $^{230}Th$ and $^{232}Th$ were performed following GEOTRACES methods in clean laboratories of the Alfred-Wegener-Institute (AWI), (Anderson et al., 2012).

Samples were spiked with $^{229}Th$ and $^{236}U$, calibrated against the reference standard material UREM11, a material in radioactive equilibrium (Hansen et al., 1983), followed by addition of a purified Fe-carrier solution ($FeCl_3$). The next day, the pH of the samples was raised to 8.5 by adding double-distilled $NH_4OH$, to induce $Fe(OH)_3$ precipitation. After 72 h, when the $Fe(OH)_3$ had settled to the bottom of the cubitainer, the precipitate was transferred from the cubitainers to acid cleaned 1 L Teflon® bottles, after syphoning off the supernatant water. After dissolution of the sample in concentrated HCl, the pH was raised again

to 8.5 to allow the $Fe(OH)_3$ precipitate and settle. The supernatant water was siphoned into acid cleaned 50 mL Falcon® tubes the following day. The samples were then washed by centrifugation four times at 4000 rpm for 12 minutes, where the supernatant was decanted before addition of new ultrapure Milli-Q® water. Finally, the precipitation was dissolved in concentrated HCl and evaporated to a drop (>10 µL) in an acid cleaned 15 mL Savillex® beaker. After evaporation, the fractions of Pa, Th, U and Nd were separated using chromatographic columns filled with anion exchange resin (AG1X8, 100-200 mesh)

according to GEOTRACES methods (Anderson et al., 2012). All fractions were collected in acid cleaned 15 mL Savillex® beakers and columns were washed and conditioned before the samples were loaded onto the columns using concentrated HCl and $HNO_3$.

Procedural blanks for $^{230}Th$ and $^{232}Th$ were run with each batch of 10-15 samples. Average $^{230}Th$ and $^{232}Th$ blank corrections are 0.24 fg/kg and 0.003 pmol/L, respectively. At station 81, a sample (2000 m) was divided into two samples and resulted in

different dissolved $^{232}Th$ concentrations, probably due to Th attached to the walls of the original cubitainer. Here, an average value considering the volume amount for both parts of the divided samples was calculated.

## 2.3 Sampling and analysis of particulate $^{234}Th$ samples collected in 2015

Particulate samples were taken using in-situ pumps (McLane and Challenger Oceanic). 268 L to 860 L seawater were pumped through a 142 mm ∅, 0.45 µm pore size Supor® (polyether sulfone) filter (Anderson et al., 2012). Filters were cut aboard for

subsamples under a laminar flow hood using tweezers and scalpels. Subsamples (23 mm ∅) were dried, put on plastic mounts, covered with Mylar and aluminium foil and directly measured by beta decay counting of $^{234}Th$ ($t_{1/2} = 24.1$ days) for at least 12 h. Six months later, background measurements were performed at the AWI in Bremerhaven.

## 2.4 Model

The model of Rutgers van der Loeff et al. (2018) was used to analyze the downward propagation of a ventilation signal in the

Atlantic layer by settling particles and radioactive ingrowth. The $^{230}Th$ model is based on the reversible exchange model of Bacon and Anderson (1982) and Nozaki et al. (1981) and solved with programming language R. We first let the $^{230}Th$ model run with the base parameters as given for the Amundsen Basin in Table 1 of Rutgers van der Loeff et al. (2018), but without





exchange with the Kara Sea, until dissolved $^{230}$Th reaches a linear steady state profile. We then simulate a ventilation of the intermediate water by introducing an exchange process down to 1500 m. A $^{230}$Th-free water mass is initially used to allow a rapid reduction of $^{230}$Th in this upper layer. The $^{230}$Th profile is determined over the full water column over time since the beginning of ventilation.

## 3. Results

$^{230}$Th results are expressed as unsupported excess $^{230}$Th ($^{230}$Th$_{xs}$); for simplification, hereinafter $^{230}$Th refers to $^{230}$Th$_{xs}$. Excess corrections were done following Hayes et al. (2015).

### 3.1 Dissolved $^{230}$Th in 1991, 2007 and 2015

Data obtained in 1991 by Scholten et al. (1995) constitute the baseline for the time series presented in this study (Fig. 2A). Dissolved $^{230}$Th activities increased with depth in the Makarov and Amundsen Basins (Scholten et al., 1995). $^{230}$Th concentrations in the Amundsen Basin (Sta. 173) were lower than in the Makarov Basin (Sta. 176) throughout the water column (Fig. 2A+C). The value observed at 2250 m in the Amundsen Basin (Sta.309) formed a mid-depth minimum in 1991.

In the Amundsen Basin, concentrations of dissolved $^{230}$Th increased more or less linearly with depth, with a slight minimum at 2750 m (Fig. 2A). In the Makarov Basin, dissolved $^{230}$Th concentrations were again higher compared to concentrations observed in the Eurasian basins in 2007. They increased until 3000 m depth, with a slight decrease towards the deepest sample (Sta. 328, Fig. 2C). Station 400, located at the south eastern margin of the Eurasian Basin showed lower concentrations than the open ocean stations.

$^{230}$Th concentrations in the Amundsen (Sta. 81, 117 and 125) and Makarov Basins (Sta. 96, 101 and 134) increased with depth in 2015. Concentrations in the Makarov Basin were up to three times higher than in the Amundsen Basin. The Makarov Basin data reveal significant internal differences in dissolved $^{230}$Th concentrations (Fig. 2C). The central Makarov Basin data (Sta. 101) have higher dissolved $^{230}$Th concentrations compared to stations located closer to the margins (Sta. 96 and 134).

### 3.2 Dissolved $^{232}$Th in 2007 and 2015

In general, the concentrations of dissolved $^{232}$Th from 2007 were close to concentrations observed in 2015. In 2015, dissolved $^{232}$Th concentrations observed in the Amundsen Basin showed a decreasing trend with depth. Surface concentrations were relatively high at station 117 (100 pmol/kg) and 125 (>200 pmol/kg). At station 81, dissolved $^{232}$Th showed a relatively constant depth distribution, where surface $^{232}$Th concentrations were lower compared to station 117 and 125. At stations 125 and 117 dissolved $^{232}$Th decreased as well slightly with depth, with station 117 showing a mid-depth maximum at 2000 m (Fig. 2B). 2007 values (station 309) decreased with depth until 2500 m and then slightly increase towards 4500 m.

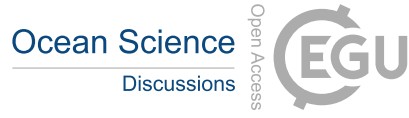

### 3.3 Particulate $^{234}$Th from 2015

Particulate $^{234}$Th from 2015 is shown as the relative amount of particulate $^{234}$Th (Fig. 2E) compared to total $^{234}$Th, calculated from $^{238}$U activities, assuming equilibrium of total $^{234}$Th with $^{238}$U in deep water (Owens et al., 2011). All profiles show rather low concentrations of particulate $^{234}$Th in the Amundsen Basin, especially below 2000 m the Nansen Basins particulate $^{234}$Th

is much higher (Valk et al., 2018).

## 4 Discussion

### 4.1 Temporal evolution of dissolved $^{230}$Th in the Amundsen Basin

Figure 3 shows the range of $^{230}$Th concentrations observed in 2015 and the temporal development since 1991. For the comparison with previous years, only changes exceeding the range of the 2015 dataset for the respective basin are considered

as significant temporal developments. As a second criterion, only changes that hold for at least three consecutive data points in a depth profile are considered as a significant temporal change. If two or three stations from 2015 show the same patterns of development, then that is considered a temporal basin wide change.

Temporal changes are manifest over the entire water column since 2007. With one exception, the 2015 concentration range is below 2007 and 1991 (Scholten et al., 1995). This difference is larger than the concentration range for the three 2015 profiles

(Fig. 3). The three stations from 2015 (81, 117 and 125) are distributed over a wide area of the Amundsen Basin (Fig. 1). Because all stations show lower concentrations in 2015, this points to a temporal rather than a regional variability over the entire basin. The decrease in dissolved $^{230}$Th in the Amundsen Basin started after 2007, considering the similar concentrations in the years 1991 and 2007. $^{230}$Th is known to respond to particle flux as well as ocean circulation (Anderson et al., 1983b, a). A reduction in dissolved $^{230}$Th concentrations can therefore be caused by either increased scavenging (Anderson et al., 1983b)

or by changing circulation (Anderson et al., 1983a).

### 4.2 Scavenging in the central Amundsen Basin

According to other studies, biological production in the Arctic ocean in 2015 was not higher than in 2007 (Ulfsbo et al., 2014). Therefore, the enhanced biological production in the Amundsen Basin and subsequent sinking particles as a major factor for the observed decrease can be excluded as a reason for the changing Th distributions. Enhanced scavenging by lithogenic

material at these stations can also be excluded because for all three stations from 2015, dissolved $^{232}$Th values at 1000 m are in the same range or lower than observed in 2007 (Fig. 2B). Low dissolved $^{232}$Th is taken here as an indicator of low amounts of lithogenic material. Enhanced particle loads would result in high concentrations of particulate $^{234}$Th, but only station 125 (2015), located at the slope of the Lomonosov Ridge shows relatively high values of particulate $^{234}$Th in the deep water (Fig. 2E). This feature could be explained by the resuspension of slope sediments along the Lomonosov Ridge, as no increased

scavenging was observed in the deep Amundsen Basin (Slagter et al., 2017). Slagter et al. (2017) argue that similar riverine surface influence of humic substances in the Amundsen Basin and in the Makarov Basin did not lead to increased scavenging





at depth in the Amundsen Basin, even at stations influenced by the TPD (e.g. station 125) (Slagter et al., 2017; Rutgers van der Loeff et al., 2018). This is in contrast to the Makarov Basin, where they observed a slight increase of dissolved Fe-binding organic ligand concentrations, and reduced dissolved Fe concentrations may point to more intense scavenging or lower Fe inputs (Slagter et al., 2017; Klunder et al., 2012). In addition, the high $^{232}$Th observed at the surface of station 125 points to a

notable continental component (Fig 2B), a signal that is not observed below (Fig. 2B). Hence, our observations are consistent with Slagter et al. (2017). To summarize, dissolved $^{232}$Th did generally not increase since 2007, except for station 117 at 2000 m and station 81 at 3500 m. Recent studies about Ra isotopes, Fe binding ligands, NCP estimates and the particulate data ($^{234}$Th, $^{232}$Th) do not point at enhanced particle fluxes in the central Amundsen Basin. Therefore, and putting all these different parameters together, it can be concluded that scavenging of $^{230}$Th within the Amundsen Basin is unlikely to be the primary

factor for the observed reduction between 2007 and 2015 in the Amundsen Basin.

### 4.3 500-1500 m: Intermediate Water mass changes

The decrease of dissolved $^{230}$Th at depths between 500 m and 1500 m for stations 81, 117 and 125 in the Amundsen Basin (2015) is most prominent at 1000 m, where concentrations decreased to half of the value in 2007 (Fig. 3). This depth range in the Amundsen Basin is ventilated on considerably shorter time scales than in the Nansen and Makarov Basin by a westward

boundary circulation (Tanhua et al., 2009).

The drop in dissolved $^{230}$Th at 1000 m corresponds to an increase in the $^{129}$I/$^{236}$U ratio (Figure 4), implying a higher Atlantic influence of younger waters (Casacuberta et al., 2018), which in turn is in agreement with an increase in the circulation/ventilation rate between 750 and 1500 m. For station 81, in the central Amundsen Basin, Rutgers van der Loeff et al. (2018) estimated a ventilation age based on SF$_6$ data of 15-18 years at 1000 m. This estimate fits to time scales based on

$^{228}$Ra data and is supported independently by the $^{129}$I/$^{236}$U ratio (Rutgers van der Loeff et al., 2018). While anthropogenic radionuclides (Fig. 4) imply exchange with young shelf waters of Atlantic influence, it is unclear to what extent the change in $^{230}$Th may be caused by exchange with the Makarov Basin. Tanhua et al. (2009) found notable changes in CFC tracer ages at the North Pole, indicating older waters in 1994 compared to 1991 and 2005 at 400 m; a change that was also documented in silicate concentrations (Tanhua et al., 2009). This feature probably reflected a shift in the front of Eurasian and Canada Basin

water around the year 1994, with Canadian Basin water penetrating deeper into the central Amundsen Basin (Tanhua et al., 2009). Unfortunately, there is no $^{230}$Th data from this phase of penetration of Canada Basin water around 1994. If the $^{230}$Th data from 1991 are connected to CFC data from the same year, while the $^{230}$Th data from 2007 are connected to CFC data of 2005 (Tanhua et al., 2009) they are both representative of periods of low intrusion of Canada Basin water over the Lomonosov Ridge. Renewed intrusion of Canada Basin water with higher dissolved $^{230}$Th concentrations in 2015 can be excluded as

mechanism for the observed change in $^{230}$Th because this would increase rather than decrease dissolved $^{230}$Th concentrations in the Amundsen Basin (Scholten et al., 1995; Edmonds et al., 2004; this study). Moreover, intrusion of Canada Basin water would not match the ventilation age estimated by Rutgers van der Loeff et al. (2018), since the Canada Basin water is known to be much older than Amundsen Basin water at this depth (Tanhua et al., 2009). Hence, it is suggested that the changes in the



Amundsen Basin cannot be explained by interaction with the Makarov Basin. On the contrary, salinity distributions imply that the influence of Atlantic waters in the Amundsen Basin has increased at 500-1500 m by 2015. Figure 2D shows salinity for three stations from the Amundsen Basin from 2007 (Schauer and Wisotzki, 2010), three from 2015 (Rabe et al., 2016), one from 1994 (Swift, 2006a) and one from 1991 (Rudels, 2010). In this depth interval the water masses shifted to notably higher

salinities in 2015, indicating that water masses have changed after 2007 (Fig. 2D). In 2015, the intermediate waters of the Amundsen Basin have a stronger Atlantic contribution (Polyakov et al., 2017; Rabe et al., 2016). This change is correlated with the decrease in dissolved $^{230}$Th.

Anthropogenic tracers can help determine whether the increased Atlantic water contribution had resulted in increased ventilation rates of the intermediate waters in the Amundsen Basin. A comparison of CFC and SF$_6$ ages between 2005 and

2015 (Fig. 5) shows that both the FSBW (approx. 425m) and the BSBW (approx. 1025m) ventilation age did not decrease after 2005. SF$_6$ age for the Atlantic Water (BSBW around 1000 m) at the northern end of the section in figure 5 is 12-15 years in 2005 and 15-18 years in 2015, suggesting perhaps a slowdown of transport of Atlantic Water in the boundary current. That would indicate that a change in scavenging along the flow path of the Atlantic water is responsible for the observed decrease in dissolved $^{230}$Th, rather than a change in ventilation.

**4.4 $^{230}$Th removal process in intermediate waters on circulation pathways**

In order to judge the scavenging intensity it is useful to compare dissolved $^{230}$Th concentrations at various locations along the flow paths of the Atlantic waters. Arctic Intermediate Water (AIW) is comprised of water from the Greenland Sea and the Nordic Sea via the West Spitzbergen Current (WSC) (Rudels, 2009). In the North East Atlantic at 25°N (GEOTRACES section GA03_W, station 20), dissolved $^{230}$Th concentrations are 8.23 fg/kg at 1000 m water depth at and 13.17 fg/kg at 1500 m (Hayes

et al., 2015) (Fig. 6). At 55°N, dissolved $^{230}$Th concentrations in 1995 were 3.47 fg/kg at 500 m and 6.8 fg/kg at 1625 m (Vogler et al., 1998) (station L3). In the Norwegian Sea, dissolved $^{230}$Th concentrations in 1993 were 5.81 fg/kg at 872 m and 7.04 fg/kg at 1286 m (Moran et al., 1995) (station 13). These values are above the highest value of dissolved $^{230}$Th at 1000 m in the Amundsen Basin in 2015 (5 fg/kg). That means that these waters have lost $^{230}$Th during their transit to the central Amundsen Basin, through the productive North Atlantic, the Fram Strait (FSBW) and over the Barents Sea shelf (BSBW).

These pathways are influenced by an increased input of terrestrial matter (Günther et al., 2013) and/or increased primary production at the shelf and the ice edge (Arrigo and van Dijken, 2015; Ulfsbo et al., 2018). Relatively high concentrations of Fe indicate the possibility of enhanced scavenging by iron oxides (Rijkenberg et al., 2018).

At station 400, located at the south eastern margin of the Eurasian Basin, the deep water is in the influence of BSBW, downstream of the Barents and Kara Sea shelf and slope. At the largest depth of ~1200m, $^{230}$Th concentration are low and

similar to concentrations in the central Amundsen Basin in 2015. This is consistent with the hypothesis that Atlantic waters that were depleted in $^{230}$Th on the shelf contribute to the decrease in dissolved $^{230}$Th in the central Amundsen Basin. Such a relic scavenging signal implies that scavenging occurs on pathways of inflow waters along the shelves rather than locally





within the central basin. The high surface values of dissolved [230]Th at station 400 are in line with low export production at this station compared to shallower stations over the shelf (Cai et al. 2010).

Hence, the observed reduction in dissolved [230]Th in the intermediate water of the Amundsen Basin is attributed to a combination of scavenging and advection. Scavenging takes place locally on the shelves and along the slopes of the Barents,

Kara and Laptev Seas, causing the removal of [230]Th observed downstream in the central Amundsen Basin. Figure 6 shows pathways of intermediate waters and dissolved [230]Th profiles from 2015, illustrating the mechanism controlling the relatively low dissolved [230]Th concentrations observed in the central Amundsen Basin. Atlantic waters flowing over the Barents and Kara shelves lose [230]Th by increased scavenging. [230]Th depleted BSBW is subducted and gradually mixes with deeper Atlantic inflow. The closer the stations are to the Lomonosov Ridge, the younger the ventilation age (Fig. 5), and the more the salinities

are shifted towards Atlantic values. Variability in temperature and salinity plots indicate that this branch interacts with ambient waters (Rudels et al., 1994). This is consistent with dissolved [230]Th concentrations observed at stations 81, 117 and 125 (2015), with station 125, located in the TPD and closest to the Lomonosov Ridge, showing the lowest concentrations. The low [230]Th concentrations at station 125 may also be affected by additional scavenging due to resuspension on the slope of the Lomonosov Ridge.

**4.5 Vertical transport of circulation derived [230]Th scavenging signal and effects in deep waters**

Increased input of Atlantic water to the central Amundsen Basin has a lower dissolved [230]Th in that depth range, due to increased scavenging during transport over the shelves and along the slope. These time series data also reveal changing conditions below the intermediate waters, indicated by a decrease of dissolved [230]Th in the deeper water column (Fig. 3).

This raises the question as to whether a change, as observed for 500-1500 m, might cause a decrease in concentrations in the

water column below that depth within just 8 years. Theoretically, such a decreasing signal could be manifest by sinking particles via reversible scavenging of sinking particles. With particle settling rates of 582 m/y (Rutgers van der Loeff et al., 2018) an average particle needs approximately six years from the depth of strongest depletion (1000 m) to reach the bottom of the water column. That would match the time scale of the decrease in [230]Th observed between 2007 and 2015. The time for particle transport to depth is the limiting step, because the time scale for particle settling is longer than for adsorption and

desorption of thorium (Rutgers van der Loeff et al., 2018). On the basis of these parameters, Rutgers van der Loeff et al. (2018) created a model to illustrate the growth of [228]Ra and [228]Th over time. This model is modified here to simulate how the full water column profile of dissolved [230]Th in the Amundsen Basin reacts to a sudden change in circulation transport of water with low [230]Th into the intermediate depth layer. The model results in figure 7 show how fast a decrease of [230]Th in the ventilated layer (500-1500 m) is propagated into the deep water. This underpins the notion of a dissolved [230]Th decrease due to circulation

and scavenging along the circulation pathways, and accounts for the reduction of dissolved [230]Th below the circulation influence within a time scale of 8 years. This temporal change can therefore be explained by a significant reduction in the input of low-[230]Th waters from shallower depths, even if the scavenging rate in the deep basin remains constant.



Hydrothermal plumes released by volcanoes at the Gakkel Ridge could also decrease dissolved [230]Th efficiently and periodically, as suggested by Valk et al. (2018) for the deep Nansen Basin. However, these plumes probably do not affect the Amundsen Basin as much as the Nansen Basin, due to recirculation in the Nansen Basin that retains most of the hydrothermal plume affected waters in the Nansen Basin (Valk et al., 2018). Additionally, the depths where the major changes occurred in the Amundsen Basin are too low (the hydrothermal scavenging starts below 2000 m) and the deep water decrease of dissolved [230]Th in the Amundsen Basin since 2007 is much weaker than in the Nansen Basin (Valk et al., 2018).

## 4.6 Development of dissolved [230]Th Makarov Basin

The change of water masses in the Amundsen Basin after 2007 could also be a result of similar changes in the Makarov Basin. Dissolved [230]Th from the Makarov Basin and temporal series are shown in figures 2C and 8, respectively. In the central Makarov Basin water mass developments are different than in the Amundsen Basin, here both salinities (above 2000 m) and dissolved [230]Th (above 1000 m) have slightly decreased since 2007 (Figure 2F). Hence the circulation changes from the Amundsen Basin did not affect directly the Makarov Basin. Theoretically, the intermediate waters of 2007 from the Amundsen Basin could have been flushed into the Makarov Basin and subsequently decreased dissolved [230]Th concentrations by mixing. The decrease of dissolved [230]Th in the intermediate waters of the Makarov Basin could also result from a stronger scavenging in the Pacific water source waters. Pacific waters enter the Arctic Ocean through the Bering Strait and undergo scavenging in the relatively high particle flux areas of the Chukchi Shelf (Vieira et al., 2018) and East Siberian Sea. These waters could reduce dissolved [230]Th concentrations in the uppermost layers of the Makarov Basin and subsequently affect deeper layers by subduction and settling particles, very similar to the scavenging process described above for the Atlantic source waters of the intermediate waters in the Amundsen Basin. Alternatively, the change can be related to other circulation changes in the Amerasian Basin for which Grenier et al. (submitted) finds evidence. These data will be discussed in Grenier et al. (submitted) in detail in the context of historical and new [230]Th data from the Canada Basin.

## 5. Conclusion

Concentrations of dissolved [230]Th throughout the entire water column in the Amundsen Basin decreased since 2007. There is no indication of increased scavenging removal of [230]Th due to particle export within the Amundsen Basin. An increase in salinity of intermediate water (at 500 - 1500m) points to the influence of Atlantic derived waters, though SF$_6$ data suggest ventilation of this layer has not increased. The reduction in dissolved [230]Th concentration in Amundsen Basin intermediate waters is therefore attributed to increased scavenging from source waters and transport of this relict scavenging signature by advection. Thus, these downstream waters reflect a scavenging history over the Siberian shelves and slope that results in a reduction of [230]Th relative to Atlantic source waters and, in turn, reduced dissolved [230]Th in the central Amundsen Basin. The low-[230]Th signal is propagated to deeper central Arctic Ocean waters by reversible scavenging. A similar reduction of [230]Th in the Makarov Basin may be related to increased scavenging over the Chukchi and East Siberian shelves. These findings



highlight the close interaction of horizontal transport by advection and particle scavenging removal, which combine to generate far-field distributions of reactive trace elements.





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



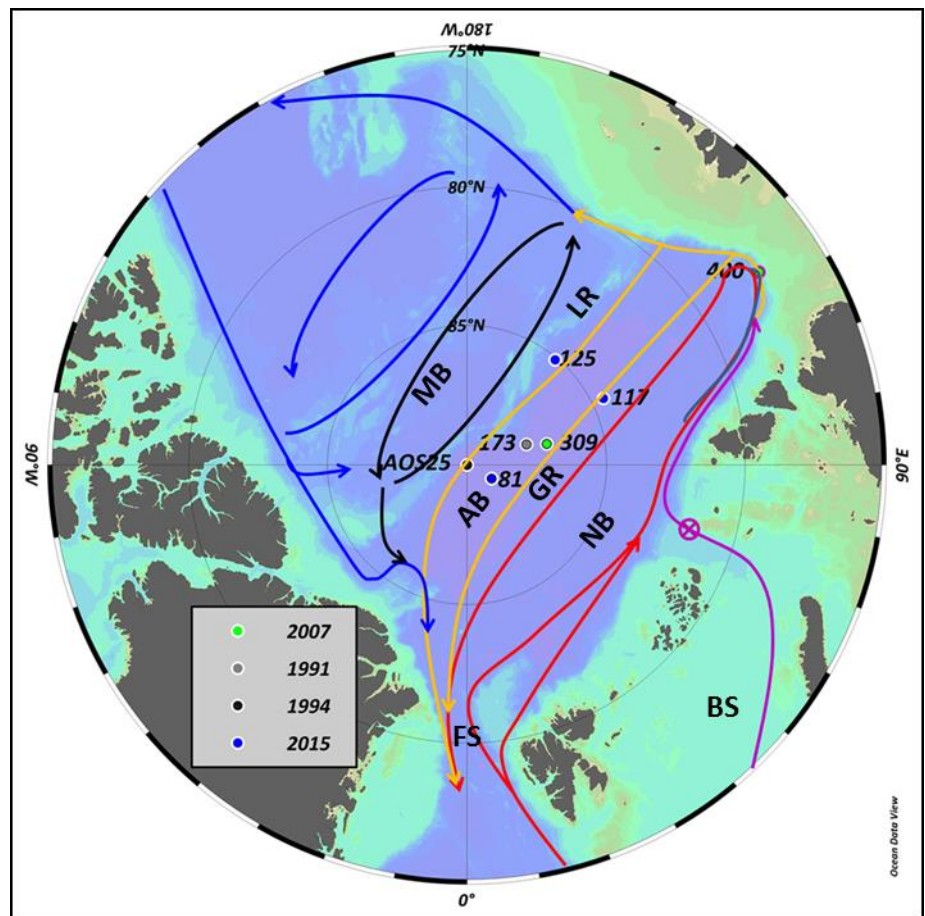

**Figure 1: Map of the Arctic Ocean and station overview. AB = Amundsen Basin, NB = Nansen Basin, MB = Makarov Basin. BS = Barents Sea, FS = Fram Strait, LR = Lomonosov Ridge with intermediate water circulation patterns after Rudels (2009). Red is the Atlantic inflow through Fram Strait (FSBW) and return flow through the Nansen Basin; purple is the inflow through the Barents Sea (BSBW). Atlantic layer circulation in the Amundsen Basin (orange), the Makarov Basin (black) and Canada Basin (blue) are indicated as arrows.**







**Figure 2: (A)** Amundsen Basin dissolved $^{230}$Th from 2015 in blue (81 = dots, 117 = squares, 125 = triangles), 2007 in green (309), and 1991 in grey (173). **(B)** Dissolved $^{232}$Th from 2015 (81 = dashed, 117 = dashed dotted, 125 = solid) and 2007 (309 = green, 400 = pink). **(C)** Makarov Basin dissolved $^{230}$Th from 2015 in orange (101 = dots, 96 = squares, 134 = triangles), 2007 in green (328), 1991 in grey (176 (Scholten et al., 1995). **(D)** Amundsen Basin salinity profiles from 2015 (Rabe et al., 2016), 2007 (Schauer and Wisotzki, 2010), 1991 (Rudels, 2010), 1994 (Swift, 2006a) and Fram Strait 2016 (Kanzow et al., 2017). **(E)** Particulate $^{234}$Th from 2015 in percent from total $^{234}$Th. **(F)** Makarov Basin salinity profiles from 2015 (Rabe et al., 2016), 2007 (Schauer and Wisotzki, 2010), 1994 (Swift, 2006b) and 1991 (Rudels, 2010).

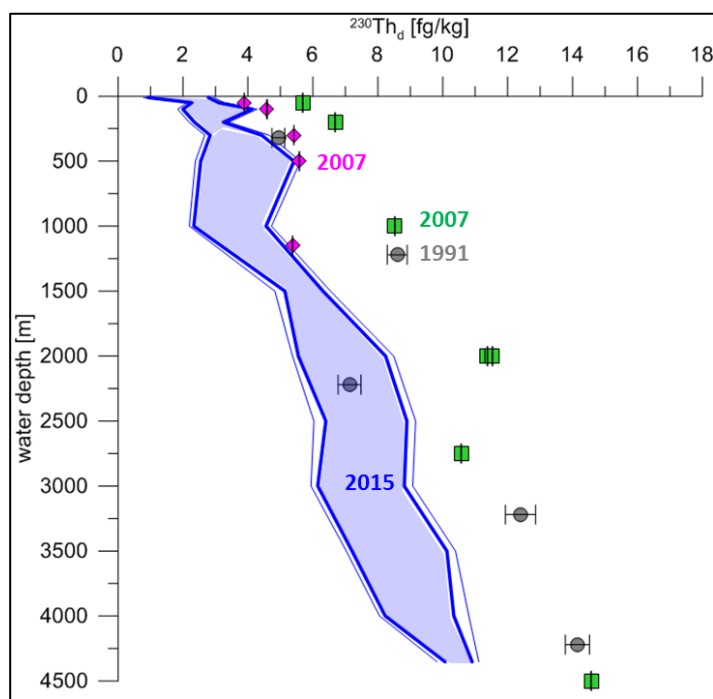

**Figure 3: Dissolved $^{230}$Th time series for the Amundsen Basin. Profiles from 2015 are combined to concentration range profiles (blue, this study, stations 81, 117, 125) and compared with data from 2007 (green, this study, station 309) and 1991 (grey, from Scholten et al. (1995) (station 176).**

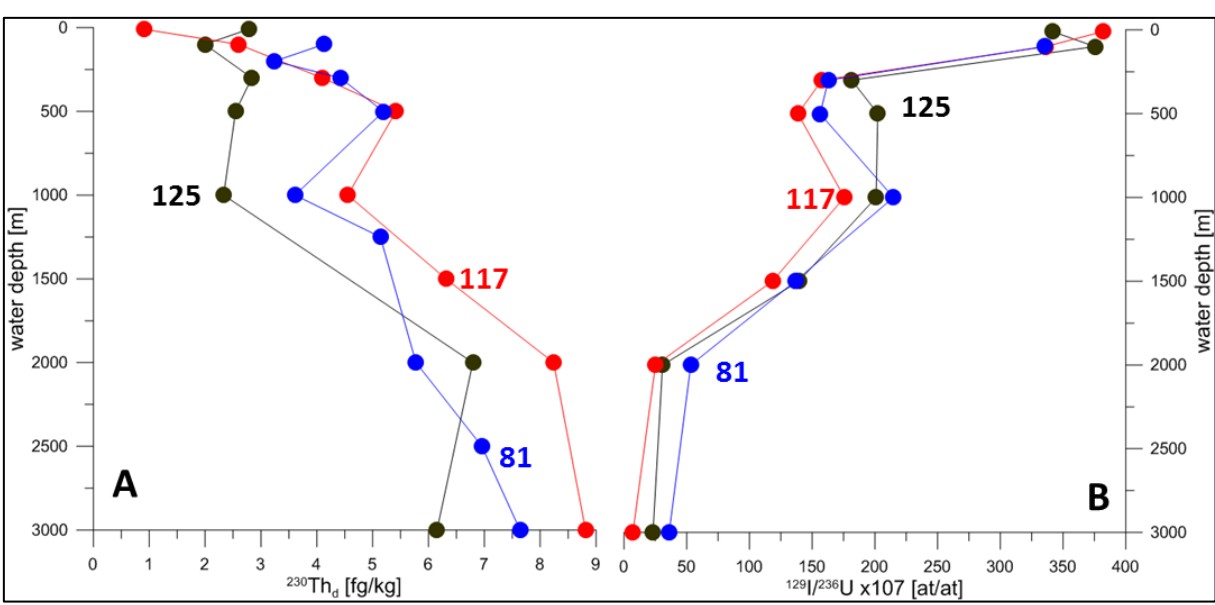

**Figure 4: (A) Dissolved $^{230}$Th and (B) $^{129}$I/$^{236}$U (Casacuberta et al., 2018) for three stations in the Amundsen Basin, 2015**

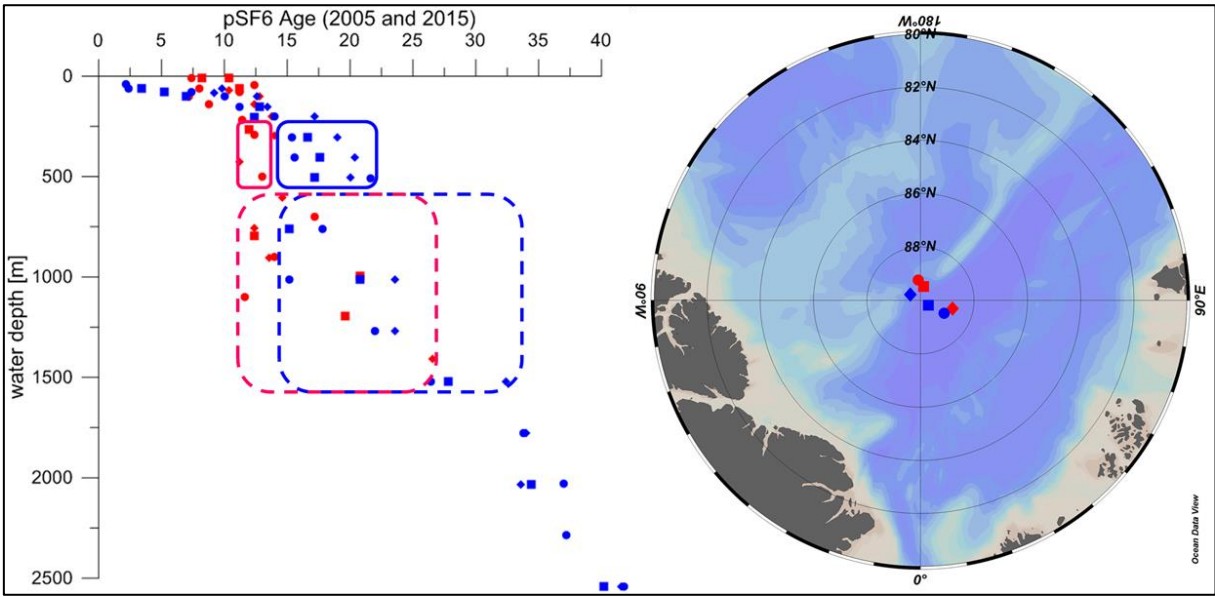

5  **Figure 5: Comparison of pCFC and pSF$_6$ ages from 2005 (red) and 2015 (blue) from the Amundsen Basin BSBW and FSBW are located in the return flow along the Lomonosov Ridge (FSBW = solid box, BSBW = dashed box). Locations of 2015 stations are marked in the map as blue symbols (81 = dots, 85 = squares, 89 = diamonds) and 2005 stations in red (41 = dots, 42 = squares, 46 = diamonds).**





**Figure 6: (A) Circulation passages of Atlantic waters to the central Amundsen Basin. (B) Conceptual drawing of scavenging and mixing of water masses close to St Anna Trough (black line in A represents the section of B). LR = Lomonosov Ridge, GR = Gakkel Ridge, BSS = Barents Sea Shelf, FS = Fram Strait). (C) Development of dissolved $^{230}$Th concentrations from the North Atlantic to the Amundsen Basin. Atlantic values: (open symbols, Hayes et al., 2015;Vogler et al., 1998;Moran et al., 1995) represented by a deep**
5 **box flowing in through Fram Strait and a shallow box with lower activities flowing in over the Barents shelf and exposed to additional scavenging on the shelf (horizontal black arrow) before it is subducted and mixed with deeper Atlantic inflow to form the observed reduced concentrations in the central Amundsen Basin. Stations 32 and 40 (red) are from Gdaniec et al. (submitted).**





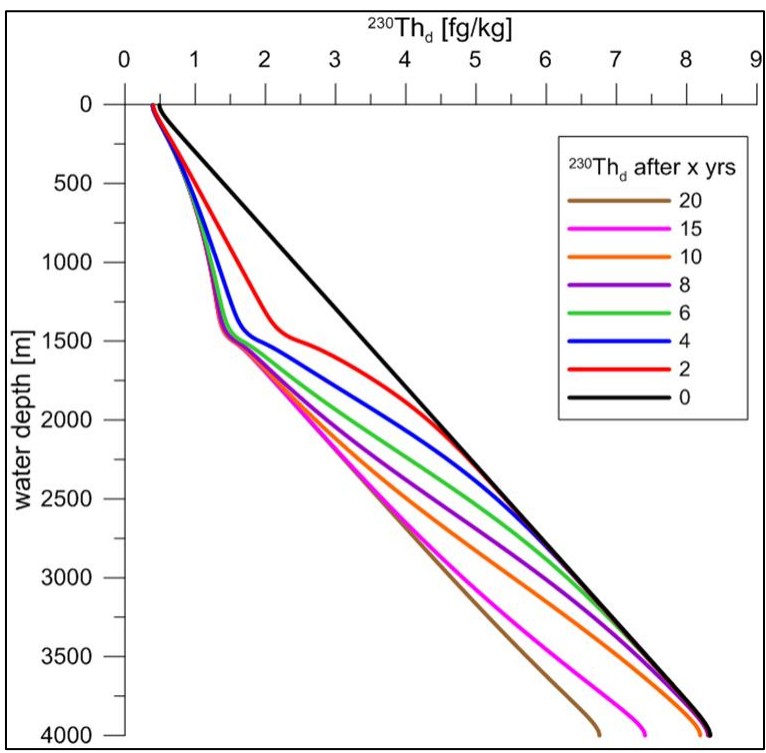

**Figure 7: Modeled dissolved $^{230}$Th distribution in the Amundsen Basin, 0, 2, 4, 6, 8, 10, 15, 20 years after reduction of concentration in upper layer (0-1500m) by continuous exchange with $^{230}$Th-free surface water. Model was modified after Rutgers van der Loeff et al. (2018).**



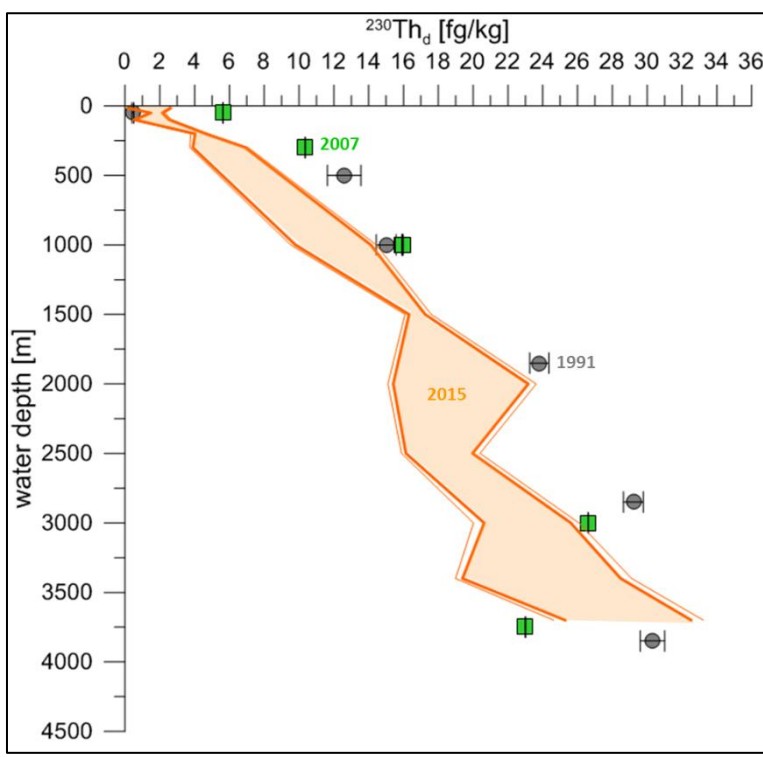

**Figure 8: Dissolved $^{230}$Th time series for the Amundsen Basin. Profiles from 2015 are combined to concentration range profiles (blue, this study, stations 81, 117, 125) and compared with data from 2007 (green, this study, station 309) and 1991 (grey, from Scholten et al. (1995) (station 176).**