# Peer review of "Decrease of dissolved 230Th in the Amundsen Basin since 2007"

_Ocean Science, 2019_

## Referee Comment (RC1) · Anonymous Referee #1 · 14 Jul 2019

The authors present a time-series study of seawater 230Th in the Arctic Ocean, which could potentially provide useful information to monitor changes in water circulation and particle dynamics in the Arctic Ocean under the impacts of climate change. The authors observed a decline in seawater 230Th in the Amundsen Basin from 1991 to 2015, which they considered to be due to the enhanced advection transporting more Atlantic water into the Arctic Ocean and increased particle scavenging during the transit. Overall, I think the study is novel and the discussion is thorough. However, I list a few issues below, which I think the authors should consider in their revision.

1. The authors suggest that the decline in seawater 230Th in the Amundsen Basin is

due to the enhanced advection of Atlantic water and the increased particle scavenging on the shelf in the Atlantic water pathway. I was wondering if the authors have any explanation why the scavenging only affects 230 Th but not 232Th in the water pathway? Another related question, if the scavenging on the shelf were the dominant signal to explain the decline in 230Th over time, one would imagine that the terrestrial signal should increase and the salinity should decrease. Why is the salinity increasing instead of decreasing in this case?

2. It would be more useful if the authors could provide a quantitative analysis to determine the rate of decline in 230Th from 1991 to 2015, and then to use other tracers (e.g. salinity or 129I/236U) to distinguish the signal of water advection from particle scavenging, so that the authors can work out the change of particle scavenging fluxes on the shelf over this period. These calculations could provide more meaningful information for ocean modeling in the Arctic Ocean.

3. I understand that the reversible exchange model used in section 2.4 and 4.5 (and Fig. 7) is cited from another reference. To help to clarify the model in this manuscript, please provide some details of the model either in the method section (if there is enough room) or in a supplementary document. In Fig.7, please add the measured data for comparison.

Minor comments: Fig.8, the caption is wrong (the current version is a repeat of Fig.3). P3/L5, The Lomonosov Ridge into the... P7/L28, ...then slightly increased towards 4500m.

---

## Referee Comment (RC2) · Anonymous Referee #2 · 16 Jul 2019

Valk et al provide convincing evidence of decreasing 230Th concentrations in the Amundsen Basin, which they attribute to increased scavenging over Eurasian shelves followed by transport of the water masses to the central Arctic. They use data from three different campaigns to show the 230Th decrease between 2007 and 2015, and they support their explanation for the observed decrease with a model. The study presents important and novel results, but certain points in the text and figures should be clarified and condensed before publication. Specific comments are listed below.

General comments:

- Section 1.3.1: Instead of reviewing the previously published studies chronologically,

it would be more helpful if similar studies were grouped together (i.e. the Moran 2005 and Hoffmann 2013 studies, which both focused on 231Pa/230Th ratios in sediments and came to similar conclusions about Pa export). This way the focus is on the results and current state of knowledge, instead of on the individual papers.

- Section 3.1: This results section is repetitive and should be condensed. Instead of listing the results by year, similar results could be stated in one sentence (e.g. "The depth profiles of 230Th increased with depth in 1991, 2007, and 2015", or, "Concentrations in the deep Makarov Basin were always higher than those observed in the Amundsen Basin")

- Sections 2.1 and 4.5: The Rutgers van der Loeff (2018) model provides support for the authors' hypothesis that the signal of increased scavenging over the shelves can be transported to the basin and propagated through the deeper parts of the water column. However, more details on the model would be appreciated in section 2.1, specifically regarding the exchange process used to introduce the ventilated water mass. Also, Figure 7 shows that after 8 years the ventilation signal does not reach below 3000 m. Can the authors comment on this difference between the model and data?

- Section 4.6: The first sentence of this section states that the processes affecting 230Th in the Makarov Basin could also provide an explanation for the changes observed in the Amundsen. The first hypothesis suggests that intermediate waters are advected from the Amundsen to the Makarov, and then the 230Th concentrations are reduced by mixing. However, the 230Th concentrations in the Makarov are higher than those in the Amundsen, so it is not clear how mixing with other Makarov basin water masses would reduce concentrations, and it is also not clear how this could provide an explanation for the decrease observed in the Amundsen Basin. Further, this section states that the salinities increased in the Makarov Basin, as opposed to the decrease observed in the Amundsen, which contradicts the first sentence of this section that suggests the changes are occurring through the same mechanism.

[Figure]

- Figures 3 and 8 present the exact same data as figures 2A and 2C. I recommend that they be removed from the manuscript.

- Title and last sentence of abstract: The title and abstract suggest that the 230Th data indicate changes in circulation, but my understanding is that the decrease in 230Th suggests increased scavenging over the shelves, not a change in circulation pathways. The changes in salinity indeed reflect a larger influence of Atlantic water, but the wording of the title suggests that the circulation changes are deduced from Th, not salinity. I suggest the authors consider a rephrasing of the title and/or abstract.

Specific comments:

- Section 1.1 heading: "patters" should be "patterns"

- Page 3, line 23 (Section 1.2): Please cite the reference that states that shelves make up 30% of the area of the Arctic. I am more familiar with the Jakobsson 2002 study (doi: 10.1029/2001GC000302) that says shelves make up >50% of the area.

- Page 4, lines 7-8 (Section 1.3): Add the half-lives of the other Th isotopes here

- Page 5, line 13 (Section 1.4): Rephrase this sentence, as written it suggests that all of these processes are declining instead of changing ("...ice cover is rapidly declining, as are changes in....")

- Page 7, line 3 (Section 3): Please briefly explain the excess corrections described in Hayes et al 2015, since this correction is important to the results presented here.

- Section 3.2: A sentence should be included to explain the pink 2007 station 400 results if this data is shown on the figure.

- Page 8, lines 4-5 (Section 3.3): I am confused by the last sentence in this section, which compares the Amundsen and Nansen Basins. Because the particulate concentrations are not shown on the figure, please state the range of particulate concentrations in question (for both the Amundsen and Nansen Basins).

- Page 9, line 6: Switch "generally" and "did" so the sentence reads "generally did not"

- Page 10: Lines 1-2 and 5-6 are both stating that intermediate waters in the Amundsen had more Atlantic influence in 2015; these two sentences should be combined to avoid repetition.

- Page 10, line 19: Delete "water depth at" so the sentence reads "...concentrations are 8.23 fg/kg at 1000 m and ..."

- Page 10, line 25: This sentence references increased inputs of terrestrial matter and increased primary production; what increase is being referred to here? An increase compared to earlier years?

- Page 10, line 27: Are the high concentrations of Fe at the margin or in the basin?

- Page 10, line 28: Instead of saying "... the deep water is in the fluence of BSBW" I suggest saying "deepest water", because the depth of 1200 m is relatively shallow compared to the other stations.

- Section 4.6: The heading for this section may be missing a word, it does not make sense as written.

- Page 12, line 9: I suggest deleting this sentence and putting the figure references at the end of the first sentence instead.

- Conclusion: I find the conclusion well-written and a good summary of the paper.

Figures:

- Figure 1: Please include stations from the Makarov Basin that are referenced in the study.

- Figure 1: The Gakkel Ridge (GR) is not defined in the caption, and is not visible with the chosen color scale.

- Figure 2: Figures 2C and 2F are from the Makarov Basin while all the other panels

are from the Amundsen. I suggest making panels 2C and 2F a separate figure, or at least putting them next to each other (by switching panels 2C and 2E).

- Figure 2: If the stations on Figure 2E are numbered, there is no need to plot them in different colors. It would be easier to follow if they were all blue (to indicate data from 2015), and the different symbols used to denote the station numbers in panels A and B could be continued in this panel.

- Figure 2: It is confusing that one station from 2007 is plotted in pink and another in green. The pink data were collected close to the margin, while the green station was in the Amundsen Basin. The pink data therefore make it difficult to discern the trend of decreasing Th in the basin that is the focus of the study, and they are also not explained in the caption for panel A. Instead of having two labels that say "2007", I suggest specifying "2007, margin" and "2007, basin" for the pink and green stations, respectively, to make the basin trend more clear.

- Figure 2D: Move the 2015 label beneath the other labels, so it is easier to find.

- Figure 2F: Are the red/orange stations from 2015? The 2015 year label is missing.

- Figure 4: The x-axis should be at the top of the plots, consistent with the other depth profile figures.

- Figure 5: I think a period is missing after the first sentence of the caption (between Amundsen Basin and BSBW)?

- Figure 6: It would be helpful to keep the symbols and colors the same between panels A and C.

- Figure 8: The caption is incorrect (should be Makarov Basin not Amundsen Basin).

[Figure]

---

## Author Comment (AC1) · 6 Sep 2019

The authors present a time-series study of seawater 230Th in the Arctic Ocean, which could potentially provide useful information to monitor changes in water circulation and particle dynamics in the Arctic Ocean under the impacts of climate change. The authors observed a decline in seawater 230Th in the Amundsen Basin from 1991 to 2015, which they considered to be due to the enhanced advection transporting more Atlantic water into the Arctic Ocean and increased particle scavenging during the transit. Overall, I think the study is novel and the discussion is thorough. However, I list a few issues below, which I think the authors should consider in their revision.

[Figure]

1. The authors suggest that the decline in seawater 230Th in the Amundsen Basin isdue to the enhanced advection of Atlantic water and the increased particle scavenging on the shelf in the Atlantic water pathway. I was wondering if the authors have any explanation why the scavenging only affects 230Th but not 232Th in the water pathway? Another related question, if the scavenging on the shelf were the dominant signal to explain the decline in 230Th over time, one would imagine that the terrestrial signal should increase and the salinity should decrease. Why is the salinity increasing instead of decreasing in this case?

Reply: We cannot say whether indeed the advection of Atlantic water has increased. Generally we intend to change the focus of the discussion from circulation change towards increased scavenging on water pathways. If there was a stronger influence of Atlantic water through Fram Strait we would observe increased salinity. We intend to change the abstract so it is not proposing a circulation change as the main cause of the dissolved 230Th reduction but a change in scavenging intensities along circulation pathways.

We agree that the increased scavenging may be related to an increased terrestrial signal. But if this signal is due to increased erosion and resuspension by the longer ice-free season, this does not require an increased runoff and corresponding decrease in salinity. The increased particle input can lead to increased input of 232Th, not 230Th, which may offset the removal of 232Th by increased scavenging.

2. It would be more useful if the authors could provide a quantitative analysis to determine the rate of decline in 230Th from 1991 to 2015, and then to use other tracers (e.g. salinity or 129I/236U) to distinguish the signal of water advection from particle scavenging, so that the authors can work out the change of particle scavenging fluxes on the shelf over this period. These calculations could provide more meaningful information for ocean modeling in the Arctic Ocean.

Reply: We agree, this kind of calculations should help to improve the discussion about

processes decreasing the dissolved 230Th concentrations in the Amundsen Basin. We will provide rates of decrease of 230Th for the period between 1991 and 2007 and between 2007 and 2015. We use data of other tracers (129I/236U) to show that there is no reason to believe that the advection has increased over these time intervals.

3. I understand that the reversible exchange model used in section 2.4 and 4.5 (and Fig. 7) is cited from another reference. To help to clarify the model in this manuscript, please provide some details of the model either in the method section (if there is enough room) or in a supplementary document. In Fig.7, please add the measured data for comparison.

Reply: The basis of the model will be included at the end of the method section. Reviewer 2 made a similar suggestion.

Minor comments: Fig.8, the caption is wrong (the current version is a repeat of Fig.3). P3/L5, The Lomonosov Ridge into the: : : P7/L28, : : :then slightly increased towards 4500m.

Reply:This will be changed accordingly.

---

## Author Comment (AC2) · 6 Sep 2019

Valk et al provide convincing evidence of decreasing 230Th concentrations in the Amundsen Basin, which they attribute to increased scavenging over Eurasian shelves followed by transport of the water masses to the central Arctic. They use data from three different campaigns to show the 230Th decrease between 2007 and 2015, and they support their explanation for the observed decrease with a model. The study presents important and novel results, but certain points in the text and figures should be clarified and condensed before publication. Specific comments are listed below. General comments: - Section 1.3.1: Instead of reviewing the previously published studies

chronologically, it would be more helpful if similar studies were grouped together (i.e. the Moran 2005 and Hoffmann 2013 studies, which both focused on 231Pa/230Th ratios in sediments and came to similar conclusions about Pa export). This way the focus is on the results and current state of knowledge, instead of on the individual papers.

Reply: Yes we will change this organization of literature review in the introduction and we are confident that this will help to better understand the manuscript.

- Section 3.1: This results section is repetitive and should be condensed. Instead of listing the results by year, similar results could be stated in one sentence (e.g. "The depth profiles of 230Th increased with depth in 1991, 2007, and 2015", or, "Concentrations in the deep Makarov Basin were always higher than those observed in the Amundsen Basin")

Reply: The result section has been condensed to half the original length.

- Sections 2.1 and 4.5: The Rutgers van der Loeff (2018) model provides support for the authors' hypothesis that the signal of increased scavenging over the shelves can be transported to the basin and propagated through the deeper parts of the water column. However, more details on the model would be appreciated in section 2.1, specifically regarding the exchange process used to introduce the ventilated water mass. Also, Figure 7 shows that after 8 years the ventilation signal does not reach below 3000 m. Can the authors comment on this difference between the model and data?

Reply: A description of how the model works will be added to the methods section. The model assumptions, such as particle sinking speed and exchange between dissolved and particulate phases might cause a difference between model and data, due to uncertainties. This may explain why the downward penetration of the ventilation signal is slower in the model than in the observed data.

- Section 4.6: The first sentence of this section states that the processes affecting 230Th in the Makarov Basin could also provide an explanation for the changes observed in the Amundsen. The first hypothesis suggests that intermediate waters are advected from the Amundsen to the Makarov, and then the 230Th concentrations are reduced by mixing. However, the 230Th concentrations in the Makarov are higher than those in the Amundsen, so it is not clear how mixing with other Makarov basin water masses would reduce concentrations, and it is also not clear how this could provide an explanation for the decrease observed in the Amundsen Basin. Further, this section states that the salinities increased in the Makarov Basin, as opposed to the decrease observed in the Amundsen, which contradicts the first sentence of this section that suggests the changes are occurring through the same mechanism.

We intend to remove the whole Makarov Basin section because it does not contribute to the discussion.

- Figures 3 and 8 present the exact same data as figures 2A and 2C. I recommend that they be removed from the manuscript.

This will be removed according to the reviewers' suggestions.

- Title and last sentence of abstract: The title and abstract suggest that the 230Th data indicate changes in circulation, but my understanding is that the decrease in 230Th suggests increased scavenging over the shelves, not a change in circulation pathways. The changes in salinity indeed reflect a larger influence of Atlantic water, but the wording of the title suggests that the circulation changes are deduced from Th, not salinity. I suggest the authors consider a rephrasing of the title and/or abstract.

We will modify the abstract and the title accordingly to avoid the impression that circulation change was concluded from 230Th data. The manuscript will focus more specifically on circulation change versus scavenging removal.

Specific comments: - Section 1.1 heading: "patters" should be "patterns" - Page 3, line 23 (Section 1.2): Please cite the reference that states that shelves make up 30% of the area of the Arctic. I am more familiar with the Jakobsson 2002 study (doi:

10.1029/2001GC000302) that says shelves make up >50% of the area.

The 30% percent statement resulted from a misunderstanding. This will be corrected.

- Page 4, lines 7-8 (Section 1.3): Add the half-lives of the other Th isotopes here

Yes that will make sense.

- Page 5, line 13 (Section 1.4): Rephrase this sentence, as written it suggests that all of these processes are declining instead of changing (": : :ice cover is rapidly declining, as are changes in: : :.")

Yes it was misleading

- Page 7, line 3 (Section 3): Please briefly explain the excess corrections described in Hayes et al 2015, since this correction is important to the results presented here.

- 230Th concentrations are corrected for a proportion of 230Th released by dissolution of lithogenic particles. This is based on parallel measurements of 232Th, considering a lithogenic ratio 230Th/232Th = $4.0 \times 10{-6}$ mol/mol (Roy-Barman et al., 2009). We will include this in the revision.

- Section 3.2: A sentence should be included to explain the pink 2007 station 400 results if this data is shown on the figure.

This will be included in the modified manuscript.

- Page 8, lines 4-5 (Section 3.3): I am confused by the last sentence in this section, which compares the Amundsen and Nansen Basins. Because the particulate concentrations are not shown on the figure, please state the range of particulate concentrations in question (for both the Amundsen and Nansen Basins).

Reply: Particulate 234Th from the Nansen Basin ranges between 1.4 and 3.3 % of total 234Th above 1500m and below they range between 3.3 and 9.1% of total 234Th.

- Page 9, line 6: Switch "generally" and "did" so the sentence reads "generally did not"

Reply: Changed

- Page 10: Lines 1-2 and 5-6 are both stating that intermediate waters in the Amundsen had more Atlantic influence in 2015; these two sentences should be combined to avoid repetition.

Reply: Changed

- Page 10, line 19: Delete "water depth at" so the sentence reads ": : :concentrations are 8.23 fg/kg at 1000 m and : : :"

Reply: Changed

- Page 10, line 25: This sentence references increased inputs of terrestrial matter and increased primary production; what increase is being referred to here? An increase compared to earlier years?

Reply: A general increase of terrestrial input compared to previous years is mentioned here.

- Page 10, line 27: Are the high concentrations of Fe at the margin or in the basin?

Reply: Concentrations at the margin are meant here, this will be stated explicitly.

- Page 10, line 28: Instead of saying ": : : the deep water is in the fluence of BSBW" I suggest saying "deepest water", because the depth of 1200 m is relatively shallow compared to the other stations.

Reply: We agree and change the text accordingly.

- Section 4.6: The heading for this section may be missing a word, it does not make sense as written.

Reply: We think the Makarov Basin section should be removed.

- Page 12, line 9: I suggest deleting this sentence and putting the figure references at the end of the first sentence instead.

Reply: This will be changed accordingly.

- Conclusion: I find the conclusion well-written and a good summary of the paper.

Reply: Thank you.

Figures: - Figure 1: Please include stations from the Makarov Basin that are referenced in the study.

Reply: We will include those stations.

- Figure 1: The Gakkel Ridge (GR) is not defined in the caption, and is not visible with the chosen color scale.

Reply: Both have been changed now.

- Figure 2: Figures 2C and 2F are from the Makarov Basin while all the other panels are from the Amundsen. I suggest making panels 2C and 2F a separate figure, or at least putting them next to each other (by switching panels 2C and 2E).

Reply: If the Makarov Basin section had remained in the manuscript, we would have structured the figure in a more logical way as proposed, but in line with suggestions from both reviewers this section will be removed.

- Figure 2: If the stations on Figure 2E are numbered, there is no need to plot them in different colors. It would be easier to follow if they were all blue (to indicate data from 2015), and the different symbols used to denote the station numbers in panels A and B could be continued in this panel.

Reply: All 2015 stations can now be identified by their blue colour.

- Figure 2: It is confusing that one station from 2007 is plotted in pink and another in green. The pink data were collected close to the margin, while the green station was in the Amundsen Basin. The pink data therefore make it difficult to discern the trend of decreasing Th in the basin that is the focus of the study, and they are also

not explained in the caption for panel A. Instead of having two labels that say "2007", I suggest specifying "2007, margin" and "2007, basin" for the pink and green stations, respectively, to make the basin trend more clear.

Reply: The label has now been changed to 2007 margin station.

- Figure 2D: Move the 2015 label beneath the other labels, so it is easier to find.

Reply: Changed

- Figure 2F: Are the red/orange stations from 2015? The 2015 year label is missing.

Yes they are. The label has been added.

- Figure 4: The x-axis should be at the top of the plots, consistent with the other depth profile figures.

Reply: We changed it accordingly.

- Figure 5: I think a period is missing after the first sentence of the caption (between Amundsen Basin and BSBW)?

Reply: Yes it was missing and has been inserted now.

- Figure 6: It would be helpful to keep the symbols and colors the same between panels A and C.

Reply: Yes, we agree and will change the colours accordingly.

- Figure 8: The caption is incorrect (should be Makarov Basin not Amundsen Basin). Interactive comment on Ocean Sci. Discuss., https://doi.org/10.5194/os-2019-49, 2019.

Reply:, that was wrong and has been changed now.

---

## Author Response (AR1)

The authors present a time-series study of seawater 230Th in the Arctic Ocean, which could potentially provide useful information to monitor changes in water circulation and particle dynamics in the Arctic Ocean under the impacts of climate change. The authors observed a decline in seawater 230Th in the Amundsen Basin from 1991 to 2015, which they considered to be due to the enhanced advection transporting more Atlantic water into the Arctic Ocean and increased particle scavenging during the transit. Overall, I think the study is novel and the discussion is thorough. However, I list a few issues below, which I think the authors should consider in their revision.

1. The authors suggest that the decline in seawater 230Th in the Amundsen Basin Is due to the enhanced advection of Atlantic water and the increased particle scavenging on the shelf in the Atlantic water pathway. I was wondering if the authors have any explanation why the scavenging only affects 230Th but not 232Th in the water pathway? Another related question, if the scavenging on the shelf were the dominant signal to explain the decline in 230Th over time, one would imagine that the terrestrial signal should increase and the salinity should decrease. Why is the salinity increasing instead of decreasing in this case?

*Reply: We cannot say whether indeed the advection of Atlantic water has increased. Generally we have changed the focus of the discussion from circulation change towards increased scavenging on water pathways. We have changed the abstract so it is not proposing a circulation change as the main cause of the dissolved 230Th reduction but a change in scavenging intensities along circulation pathways.*
*We agree that the increased scavenging may be related to an increased terrestrial signal. But if this signal is due to higher temperature, increased erosion and resuspension by the longer ice-free season, this does not require an increased runoff and corresponding decrease in salinity. The increased particle input can lead to increased input of 232Th, not 230Th, which may offset the removal of 232Th by increased scavenging.*

2. It would be more useful if the authors could provide a quantitative analysis to determine the rate of decline in 230Th from 1991 to 2015, and then to use other tracers (e.g. salinity or 129I/236U) to distinguish the signal of water advection from particle scavenging, so that the authors can work out the change of particle scavenging fluxes on the shelf over this period. These calculations could provide more meaningful information for ocean modeling in the Arctic Ocean.

*Reply: We now argue based on atmospherically derived tracers (CFC, SF6) that the ventilation rate of the intermediate water in the Amundsen Basin has not increased over the investigated period. We therefore explain the decrease of total $^{230}$Th in this water mass as entirely due to increased scavenging on the shelf. We can make a back-of-the-envelope estimate of the required scavenging flux. The inflow of Atlantic Water through the Barents Sea is about 1.5 Sv (Ingvaldsen, R.B., Asplin, L., Loeng, H., 2004. The seasonal cycle in the Atlantic transport to the Barents Sea during the years 1997–2001. Continental Shelf Research 24, 1015-1032) with a $^{230}$Th concentration of 3 fg/kg (Fig. 5 of manuscript) or 145 g $^{230}$Th/yr. The observed reduction of 4 fg/kg $^{230}$Th in the upper 1500m of the Amundsen Basin (250 x 1000km) amounts to a removal of 1500g 230Th. The observed removal in the Amundsen Basin would require removal of all $^{230}$Th from the Atlantic inflow over 11 years or an equivalent amount in waters flowing along the Barents Sea slope. This calculation shows that the process is of the right order of magnitude, but we feel it is too crude to include it in the manuscript.*

3. I understand that the reversible exchange model used in section 2.4 and 4.5 (and
Fig. 7) is cited from another reference. To help to clarify the model in this manuscript,
please provide some details of the model either in the method section (if there is
enough room) or in a supplementary document. In Fig.7, please add the measured
data for comparison.
*Reply: Reviewer 2 made a similar suggestion. We have now included a table that describes*
*the essential model parameters.*
Minor comments: Fig.8, the caption is wrong (the current version is a repeat of Fig.3).
P3/L5, The Lomonosov Ridge into the: : : P7/L28, : : :then slightly increased towards
4500m.
*Figure 8 has been removed*

Valk et al provide convincing evidence of decreasing 230Th concentrations in the
Amundsen Basin, which they attribute to increased scavenging over Eurasian shelves
followed by transport of the water masses to the central Arctic. They use data from
three different campaigns to show the 230Th decrease between 2007 and 2015, and
they support their explanation for the observed decrease with a model. The study
presents important and novel results, but certain points in the text and figures should be
clarified and condensed before publication. Specific comments are listed below. General
comments: - Section 1.3.1: Instead of reviewing the previously published studies
chronologically, it would be more helpful if similar studies were grouped together (i.e.
the Moran 2005 and Hoffmann 2013 studies, which both focused on 231Pa/230Th ratios
in sediments and came to similar conclusions about Pa export). This way the focus
is on the results and current state of knowledge, instead of on the individual papers.
*Reply: The review in section 1.3.1 has been changed in line with the reviewers suggestions.*
- Section 3.1: This results section is repetitive and should be condensed. Instead of
listing the results by year, similar results could be stated in one sentence (e.g. "The
depth profiles of 230Th increased with depth in 1991, 2007, and 2015", or, "Concentrations
in the deep Makarov Basin were always higher than those observed in the
Amundsen Basin")
*Reply: The result section has been condensed.*
- Sections 2.1 and 4.5: The Rutgers van der Loeff (2018) model provides support for
the authors' hypothesis that the signal of increased scavenging over the shelves can be
transported to the basin and propagated through the deeper parts of the water column.
However, more details on the model would be appreciated in section 2.1, specifically
regarding the exchange process used to introduce the ventilated water mass. Also,
Figure 7 shows that after 8 years the ventilation signal does not reach below 3000 m.
Can the authors comment on this difference between the model and data?
*Reply: A table with the critical model parameters has been added to the methods section.*
*The exchange process used to introduce the ventilated water mass is not meant to*
*reproduce the actual ventilation with water from Kara/Barents Seas, but merely serves the*
*purpose to create a rapid reduction of $^{230}$Th in the upper 1500m in order to model the*
*downward propagation of such a signal by reversible scavenging. This has now been*
*described more clearly in the end of this section (now section 2.4).The model assumptions,*
*such as particle sinking speed and exchange between dissolved*
*and particulate phases might cause a difference between model and data, due to*
*uncertainties. This may explain why the downward penetration of the ventilation signal is*

*slower in the model than in the observed data. The model should be seen as a description of*
*the removal process that proceeds downwards, rather than a precise retrace of profiles from*
*the central Amundsen Basin. We have explained this explicitly in the revised manuscript.*
- Section 4.6: The first sentence of this section states that the processes affecting
230Th in the Makarov Basin could also provide an explanation for the changes ob-served in
the Amundsen. The first hypothesis suggests that intermediate waters are
advected from the Amundsen to the Makarov, and then the 230Th concentrations are
reduced by mixing. However, the 230Th concentrations in the Makarov are higher than
those in the Amundsen, so it is not clear how mixing with other Makarov basin water
masses would reduce concentrations, and it is also not clear how this could provide an
explanation for the decrease observed in the Amundsen Basin. Further, this section
states that the salinities increased in the Makarov Basin, as opposed to the decrease
observed in the Amundsen, which contradicts the first sentence of this section that
suggests the changes are occurring through the same mechanism.
*The whole Makarov Basin section has been removed because it did not contribute*
*to the discussion.*
- Figures 3 and 8 present the exact same data as figures 2A and 2C. I recommend that
they be removed from the manuscript.
*This has been removed according to the reviewers' suggestions.*
- Title and last sentence of abstract: The title and abstract suggest that the 230Th data
indicate changes in circulation, but my understanding is that the decrease in 230Th
suggests increased scavenging over the shelves, not a change in circulation pathways.
The changes in salinity indeed reflect a larger influence of Atlantic water, but the wording
of the title suggests that the circulation changes are deduced from Th, not salinity.
I suggest the authors consider a rephrasing of the title and/or abstract.
*We have changed the abstract and the title accordingly to avoid the impression that*
*circulation change was concluded from 230Th data.*
Specific comments: - Section 1.1 heading: "patters" should be "patterns" - Page 3,
line 23 (Section 1.2): Please cite the reference that states that shelves make up 30%
of the area of the Arctic. I am more familiar with the Jakobsson 2002 study (doi:
10.1029/2001GC000302) that says shelves make up >50% of the area.
*The 30% percent statement resulted from a misunderstanding. This has been corrected.*
- Page 4, lines 7-8 (Section 1.3): Add the half-lives of the other Th isotopes here
*The half-lives have been added.*
- Page 5, line 13 (Section 1.4): Rephrase this sentence, as written it suggests that all
of these processes are declining instead of changing (": : :ice cover is rapidly declining,
as are changes in: : :.")
*This has been changed accordingly*
- Page 7, line 3 (Section 3): Please briefly explain the excess corrections described in
Hayes et al 2015, since this correction is important to the results presented here.
*- 230Th concentrations are corrected for a proportion of 230Th released by dissolution*
*of lithogenic particles. This is based on parallel measurements of 232Th, considering*
*a lithogenic ratio 230Th/232Th = 4.0_10□6 mol/mol (Roy-Barman et al., 2009). We*
*included this in the revision.*
- Section 3.2: A sentence should be included to explain the pink 2007 station 400
results if this data is shown on the figure.
*This was included in the modified manuscript.*
- Page 8, lines 4-5 (Section 3.3): I am confused by the last sentence in this section,
which compares the Amundsen and Nansen Basins. Because the particulate concentrations
are not shown on the figure, please state the range of particulate concentrations
in question (for both the Amundsen and Nansen Basins).
*Reply: Particulate 234Th is similar in the Nansen and Amundsen Basins in the upper 1500m,*
*but in the deep Nansen Basin we find higher $^{230}$Th, in the range between 3.3 and 9.1% of*
*total 234Th.We have now specified the comparison to the **deep** Nansen Basin.*
- Page 9, line 6: Switch "generally" and "did" so the sentence reads "generally did not" Reply:
*Changed*

- Page 10: Lines 1-2 and 5-6 are both stating that intermediate waters in the Amundsen
had more Atlantic influence in 2015; these two sentences should be combined to avoid
repetition.
*Reply: Changed*
- Page 10, line 19: Delete "water depth at" so the sentence reads ": : :concentrations
are 8.23 fg/kg at 1000 m and : : :"
*Reply: Changed*
- Page 10, line 25: This sentence references increased inputs of terrestrial matter and
increased primary production; what increase is being referred to here? An increase
compared to earlier years?
*Reply: A general increase of terrestrial input compared to previous years is meant*
*here.*
- Page 10, line 27: Are the high concentrations of Fe at the margin or in the basin?
*Reply: Concentrations at the margin are meant here, this is now stated explicitly.*
- Page 10, line 28: Instead of saying ": : : the deep water is in the fluence of BSBW"
I suggest saying "deepest water", because the depth of 1200 m is relatively shallow
compared to the other stations.
*Reply: We agree and changed the text accordingly.*
- Section 4.6: The heading for this section may be missing a word, it does not make
sense as written.
*Reply: We have removed the Makarov Basin section completely.*
- Page 12, line 9: I suggest deleting this sentence and putting the figure references at
the end of the first sentence instead.
*Reply: This has been changed accordingly.*
- Conclusion: I find the conclusion well-written and a good summary of the paper.
*Reply: Thank you.*
Figures: - Figure 1: Please include stations from the Makarov Basin that are referenced
in the study.
*Reply: Description of the Makarov Basin Stations was taken out of the manuscript, so there*
*was no need to include them in the map.*
- Figure 1: The Gakkel Ridge (GR) is not defined in the caption, and is not visible with
the chosen color scale.
*Reply: Both have been changed now.*
- Figure 2: Figures 2C and 2F are from the Makarov Basin while all the other panels
are from the Amundsen. I suggest making panels 2C and 2F a separate figure, or at
least putting them next to each other (by switching panels 2C and 2E).
*Reply: If the Makarov Basin section had remained in the manuscript, we would have*
*structured the figure in a more logical way as proposed, but in line with suggestions*
*from both reviewers this section and the corresponding panels of the figure were removed.*
- Figure 2: If the stations on Figure 2E are numbered, there is no need to plot them in
different colors. It would be easier to follow if they were all blue (to indicate data from
2015), and the different symbols used to denote the station numbers in panels A and
B could be continued in this panel.
*Reply: All 2015 stations can now be identified by their blue colour.*
- Figure 2: It is confusing that one station from 2007 is plotted in pink and another
in green. The pink data were collected close to the margin, while the green station
was in the Amundsen Basin. The pink data therefore make it difficult to discern the
trend of decreasing Th in the basin that is the focus of the study, and they are also
not explained in the caption for panel A. Instead of having two labels that say "2007",
I suggest specifying "2007, margin" and "2007, basin" for the pink and green stations,
respectively, to make the basin trend more clear.
*Reply: The label has now been changed to "2007 margin" station and this station is now*
*mentioned in the caption of panel A.*
- Figure 2D: Move the 2015 label beneath the other labels, so it is easier to find.

*Reply: This is now panel B. We prefer to keep the labels unchanged, because the 2015 label*
*is now close to all the bluish profiles where the 2015 data diverge most clearly from profiles*
*of earlier expeditions.*
- Figure 2F: Are the red/orange stations from 2015? The 2015 year label is missing.
*This panel has been removed*
- Figure 4: The x-axis should be at the top of the plots, consistent with the other depth
profile figures.
*Reply: We changed it accordingly.*
- Figure 5: I think a period is missing after the first sentence of the caption (between
Amundsen Basin and BSBW)?
*Reply: The sentence has been reformulated*
- Figure 6: It would be helpful to keep the symbols and colors the same between panels
A and C.
*Reply: Yes, we agree and have changed the colours and symbols accordingly.*
- Figure 8: The caption is incorrect (should be Makarov Basin not Amundsen Basin).
*Reply: This figure has been removed*

[revised manuscript text omitted]